# DeepSpeed4Science Initiative: Enabling Large-Scale Scientific Discovery through Sophisticated AI System Technologies

**Shuaiwen Leon Song**
Microsoft

**Bonnie Kruft**
Microsoft

**Minjia Zhang**
Microsoft

**Conglong Li**
Microsoft

**Shiyang Chen**
Rutgers University

**Chengming Zhang**
Microsoft

**Masahiro Tanaka**
Microsoft

**Xiaoxia Wu**
Microsoft

**Mohammed AlQuraishi**
Columbia University

**Gustaf Ahdritz**
Harvard University

**Christina Floristean**
Columbia University

**Rick Stevens**
Argonne National Laboratory

**Venkatram Vishwanath**
Argonne National Laboratory

**Arvind Ramanathan**
Argonne National Laboratory

**Sam Foreman**
Argonne National Laboratory

**Kyle Hippe**
Argonne National Laboratory

**Prasanna Balaprakash**
Oak Ridge National Laboratory

**Yuxiong He**
Microsoft

## Abstract

In the upcoming decade, deep learning may revolutionize the natural sciences, enhancing our capacity to model and predict natural occurrences. This could herald a new era of scientific exploration, bringing significant advancements across sectors from drug development to renewable energy. To answer this call, we present DeepSpeed4Science initiative (deepspeed4science.ai) which aims to build unique capabilities through AI system technology innovations to help domain experts to unlock today's biggest science mysteries. By leveraging DeepSpeed's current technology pillars (training, inference and compression) as base technology enablers, DeepSpeed4Science will create a new set of AI system technologies tailored for accelerating scientific discoveries by addressing their unique complexity beyond the common technical approaches used for accelerating generic large language models (LLMs). In this paper, we showcase the early progress we made with DeepSpeed4Science in addressing two of the critical system challenges in structural biology research.

## 1 Introduction

In the next decade, deep learning may revolutionize the natural sciences, enhancing our capacity to model and predict natural occurrences. This could herald a new era of scientific exploration, bringing significant advancements across sectors from drug development to renewable energy. In line with Microsoft's mission to empower every person and every organization on the planet to achieve more, the DeepSpeed team at Microsoft is responding to this opportunity by launching a new initiative

---

An extended version of this paper is at arxiv.org/abs/2310.04610.

NeurIPS 2023 AI for Science Workshop.

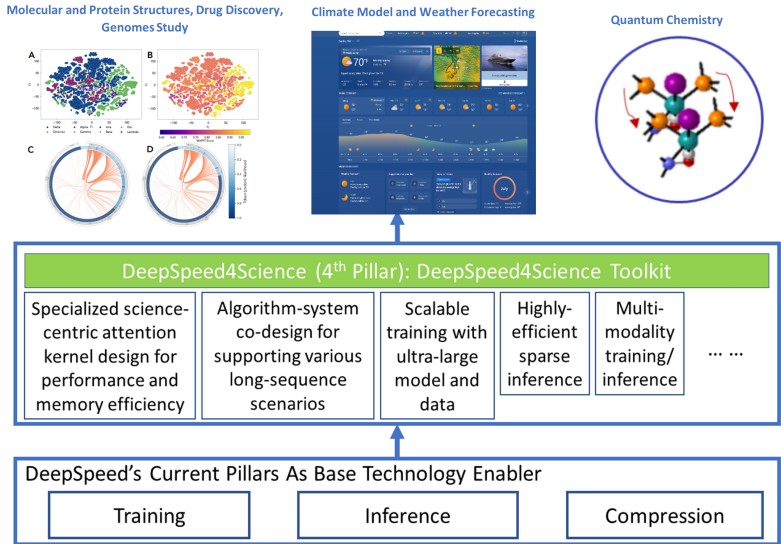

Figure 1: DeepSpeed4Science approach: developing a new set of AI system technologies that are beyond generic large language model support, tailored for accelerating scientific discoveries and addressing their complexity.

called called DeepSpeed4Science, aiming to build unique capabilities through AI system technology innovations to help domain experts to unlock today's biggest science mysteries.

The DeepSpeed system framework [9] is an industry leading open-source AI system framework, developed by Microsoft, that enables unprecedented scale and speed for deep learning training and inference on a wide range of AI hardware. Figure 1 demonstrates our basic approach to this new initiative. By leveraging DeepSpeed's current technology pillars (training, inference and compression) as base technology enablers, DeepSpeed4Science will create a new set of AI system technologies tailored for accelerating scientific discoveries by addressing their unique complexity beyond the common technical approaches used for accelerating generic large language models (LLMs). We work closely with internal and external teams who own AI-driven science models that represent key science missions, to identify and address general domain-specific AI system challenges. This includes climate science, drug design, biological understanding, molecular dynamics simulation, cancer diagnosis and surveillance, catalyst/material discovery, and other domains.

Our long-term vision is to develop DeepSpeed4Science into a new platform and a unified repository for sharing advanced AI system technologies that support scientific discoveries. DeepSpeed4Science is designed to be inclusive, which is reflected in the initiative's support for a diverse group of signature science models, representing some of the most critical AI for science investments. In this paper, we showcase how DeepSpeed4Science helps address two of their critical system challenges in structural biology research: (1) eliminating memory explosion problems for scaling Evoformer-centric protein-structure prediction models, and (2) enabling very-long sequence support for better understanding the evolutionary landscape of pandemic-causing viruses.

## 2 DS4Sci_EvoformerAttention: eliminating memory explosion problems for scaling *Evoformer-centric* structural biology models

### 2.1 Core Problem Description

OpenFold [1] is a community reproduction of DeepMind's AlphaFold2 [5] that makes it possible to train or finetune AlphaFold2 on new datasets. Researchers have used it to retrain AlphaFold2 from scratch to produce new sets of model parameters, studied the early training phase of AlphaFold2, and developed new protein folding systems.

While OpenFold does apply performance and memory optimizations using state-of-the-art system technologies, training AlphaFold2 from scratch is still computationally expensive. The model at the current stage is small in absolute terms, with just 93 million parameters, but it contains several custom attention variants that manifest unusually large activations. During the "finetuning" phase of a standard AlphaFold2 training run, the logit tensor produced in just one of these variants–one

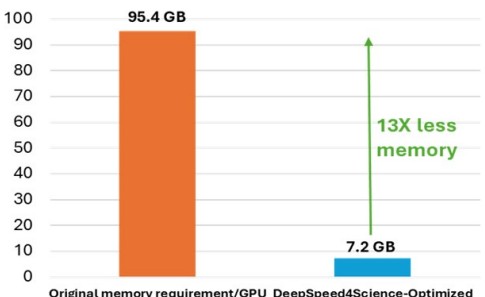

Figure 2: Peak memory requirement for training variants of the MSA attention kernels (with bias) with the maximum possible training sample dimension in OpenFold. (Left) The original OpenFold implementation with EvoformerAttention used in AlphaFold2. The memory explosion problems in training/inference these types of protein structure prediction models are common. Particularly, state-of-the-art FlashAttention cannot effectively support such science attention variants. (Right) A new solution from DeepSpeed4Science called DS4Sci_EvoformerAttention significantly reduces OpenFold's peak memory requirement for training by 13X without accuracy loss.



Figure 3: Peak memory requirement breakdown for training variants of the MSA attention kernels (with bias) with the maximum possible training sample dimension in Open-Fold. (Left bar) the original OpenFold implementation with EvoformerAttention used in AlphaFold 2. The memory explosion problems in training/inference these types of protein structure prediction models are common. Particularly, FlashAttention cannot effectively support such science attention variants. (Right bar) Our DeepSpeed4Science-optimized solution significantly reduces the overall peak memory requirement.

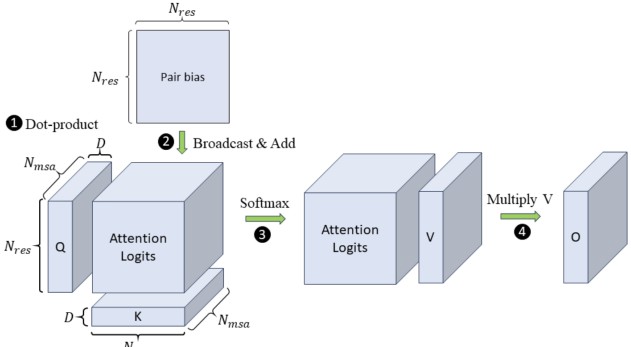

Figure 4: The example of MSA row-wise attention computation in OpenFold in four steps. The example shows the computation of one attention head, where the input Q, K, and V are 3D tensors and the pair bias is a matrix. Each attention head is associated with a 3D intermediate attention logit causing the memory explosion. We fuse four steps in one kernel to reduce peak memory usage.

designed to attend over the deep protein MSAs fed to the model as input–is in excess of 12GB in half precision alone, dwarfing the peak memory requirements of comparably sized language models. Even with techniques like activation checkpointing and DeepSpeed ZeRO optimizations [8], this memory explosion problem heavily constrains the sequence lengths and MSA depths on which the model can be trained. Furthermore, approximation strategies can significantly affect the model accuracy and convergence, while still resulting in memory explosion, shown as the left bar (orange) in Figure 2.

To address this common system challenge in structural biology research (e.g., protein structure prediction and equilibrium distribution prediction), DeepSpeed4Science is addressing this memory inefficiency problem by designing customized exact attention kernels for the attention variants (i.e., EvoformerAttention), which widely appear in this category of science models. Specifically, a set of highly memory-efficient DS4Sci_EvoformerAttention kernels enabled by sophisticated fusion/tiling strategies and on-the-fly memory reduction methods, are created for the broader community as high-quality machine learning primitives. Incorporated into OpenFold, they provide a substantial speedup during training and dramatically reduce the model's peak memory requirement for training

and inference. This allows OpenFold to be experimented with bigger and more complex models, and longer sequences, and trained on a wider spectrum of hardware.

## 2.2 Methodology

The Evoformer-centric models such as OpenFold and others typically use four attention variants to process the 4D sequence tensors: MSA row-wise, MSA column-wise, and two kinds of Triangular. In particular, the input tensor is of shape $(N_{res}, N_{msa}, H, D)$, where $N_{msa}$ is the length of MSA sequences, $N_{res}$ is the length of residue sequences, $H$ is the number of attention heads, and $D$ is the hidden dimension of the model. Figure 4 illustrates an example of MSA row-wise attention. The inputs consist of three projected tensors in shape $(N_{msa}, N_{res}, D)$, namely $Q$, $K$, and $V$, and a $(N_{res}, N_{res})$ bias matrix of residue pairs. In step 1, $Q$ and $K$ perform dot-product between every row vector along the $D$ dimension, deriving the attention logits in shape $(H, N_{msa}, N_{res}, N_{res})$ as the intermediate results. For simplicity, we only depict one head in the figure. Unlike language models such as GPT-3 [2], where $D$ and $H$ are considerably larger, Evoformer operates on a different scale. Specifically, MSA row-wise attention is typically designed with 8 heads, each having 8 features, while GPT-3 is configured with 96 heads and 128 features per head. However, MSA and residue sequence lengths can extend up to 5K during training and inference, respectively, making the memory explosion for intermediate results. MSA row-wise attention has the $O(N_{msa} * N_{res}^2)$ memory footprint, and, similarly, for MSA column-wise attention, the memory footprint is $O(N_{res} * N_{msa}^2)$. In contrast, the memory footprint of language models is much smaller, approximately $O(N^2)$. Figure 3 shows the breakdown of memory requirements per GPU.

Existing techniques for long sequences cannot effectively address such memory explosion challenges in Evoformer's specialized attention for structural biology. For example, MSA row-wise attention and two Triangular attention apply a bias term to the attention logits, and the bias term's gradients are required during backward. As shown in step 2, the pair bias is derived by projecting the pair-wise representation and is used to adjust the attention logits based on the structure of residues to satisfy the spatial constraints. Take FlashAttention [4] as an example; it cannot integrate these backward-compatible bias terms directly. Furthermore, the bias requires appropriate broadcasting to match the shape of attention logits before adding. It thus also needs to be mirrored in backward computing. Recognizing these challenges, DeepSpeed4Science addresses this memory inefficiency problem by designing customized, exact attention kernels for these attention variants in EvoformerAttention and boosting the training/inference efficiency.

Our customized highly memory-efficient DS4Sci_EvoformerAttention kernels fuses the four steps computation and calculates the attention logits in tiles. Specifically, in the forward kernel, each thread block computes a tile of $(Tile_x, Tile_y, Tile_z)$ in the attention logit tensor. Each thread block loads the needed tiles from $Q$ and $K$ to perform the dot-product. The resultant tile is stored in registers and added with biases. Then, we perform softmax as step 3 and multiply $V$ as step 4. We reduce the memory footprint by materializing only a subset of tiles in the $(N_{msa}, N_{res}, N_{res})$ tensor and not saving the whole tensor for backward. We perform steps 1-3 in the backward kernel to recompute the attention logits. The backward computation is similar to that of FlashAttention. In our kernels, we tune the tile size for better performance. Large tile size leads to more efficient memory access while incurring register spilling; We tune the tile size to be $(64, 64, 1)$.

The bias-adding needs to be effectively broadcasted to match the bias shape with the attention logits. For example, in MSA row-wise attention, the residue pair-wise representation in shape $(N_{msa}, N_{res}, D)$ is transformed to be the bias term in shape $(H, N_{msa}, N_{res})$, while the attention logits tensor is of shape $(H, N_{msa}, N_{res}, N_{res})$. To broadcast, the bias tensor will be repeated $N_{msa}$ times as the second dimension. Here, we cannot directly leverage the broadcast semantics in Pytorch because we use a fused CUDA kernel out of PyTorch. Besides, broadcasting in PyTorch requires the operation between two full tensors instead of tiles. Thus, we enabled on-the-fly broadcasting in the kernel; in particular, after calculating the attention logits after step 1. For example, a thread block loads a $(Tile_x, Tile_y)$ tile from the pair bias. The thread block for different heads with the same position of its tile in the attention logits will load the same bias tile. The loaded tile is added to the logits tile in registers.

In backward, the gradient of the bias terms equals the gradient of attention logits. However, we need to reverse the broadcast operation. That is, the gradients along the broadcast dimension need to be accumulated. Specifically, the shape of attention logits gradients is $H, N_{msa}, N_{res}, N_{res}$ and the bias

gradient is computed similar to $attn\_grad.sum(0)$ in Pytorch. To reduce the memory footprint, we also fuse this operation into our kernel; otherwise, it needs the full attention logits gradient tensor. As described above, different thread blocks load the same bias tiles participants in the accumulation. Each thread block uses atomic-add operations when writing out its tile of gradients. To reduce the contention that multiple thread blocks are trying to write the same place, we schedule the thread block so that blocks executing on GPU's multiprocessors at the same wave write to different tiles. Furthermore, the accumulation could lead to potential accuracy issues due to the round-off error of low-precision arithmetic operations, especially for bfloat16. Consequently, we convert the gradient to FP32 before adding and converting it back in another kernel if necessary. It also avoids using the slow $atom.add.bf16x2$ instruction.

For detailed source code release and tutorial, please visit our release blog for $DS4Sci\_EvoformerAttention$.

## 3 DeepSpeed4Science Enables Very-Long Sequence Support via both Systematic and Algorithmic Approaches for Genome-scale Foundation Models

### 3.1 Core Problem Description

GenSLMs [12], a 2022 ACM Gordon Bell award winning genome-scale language model from Argonne National Lab, can learn the evolutionary landscape of SARS-CoV-2 (COVID-19) genomes by adapting large language models (LLMs) for genomic data. It is designed to transform how new and emergent variants of pandemic-causing viruses, especially SARS-CoV-2, are identified and classified. GenSLM represents one of the first whole genome-scale foundation models which can generalize to other prediction tasks. A good understanding of the latent space can help GenSLMs tackle new domains beyond just viral sequences and expand their ability to model bacterial pathogens and even eukaryotic organisms, e.g., to understand things such as function, pathway membership, and evolutionary relationships. To achieve this scientific goal, GenSLMs and similar models require very long sequence support for both training and inference that is beyond generic LLMs' long-sequence strategies like FlashAttention. Through DeepSpeed4Science's new designs, scientists can now build and train models with significantly longer context windows, allowing them to explore relationships that were previously inaccessible.

Despite the importance of supporting very long sequence lengths and efficient training for better understanding the genome latent space in models like GenSLMs, the existing large model training frameworks such as NVIDIA Megatron-LM [7] and old version of Megatron-DeepSpeed [6], and their corresponding parallelism choices do not have tailored optimizations for very long sequence training and inference. There are two main challenges with the existing frameworks. First, the existing parallelism approaches such as data, tensor, and pipeline parallelism cannot effectively address the scaling along the sequence dimension. Second, the existing large model training systems feature inferior training throughput when long sequences are required. For example, many scientists today use NVIDIA's Megatron-LM or the older version of Megatron-DeepSpeed to train their models. Megatron-DeepSpeed is the DeepSpeed version of NVIDIA's Megatron-LM. GenSLMs were previously trained with Megatron-DeepSpeed. However, the older version of Megatron-DeepSpeed misses many new acceleration opportunities including FlashAttention2 [3], new fused kernels and sequence parallelism. As shown in Figure 5, the maximum sequence lengths supported by the two state-of-the-art frameworks for the 33B GenSLM model are less than 60K, which is far from the requirements of the genome-scale foundation models. And even worse, they show very poor scalability in training.

In this release, we are proud to introduce the new Megatron-DeepSpeed framework. We rebased and enabled DeepSpeed with the newest Megatron for long sequence support and other capabilities/optimizations. With the new Megatron-DeepSpeed, users can now train their large AI4Science models like GenSLMS with much longer sequences via a synergetic combination of our newly added memory optimization techniques on attention mask and position embedding, tensor parallelism, pipeline parallelism, sequence parallelism, ZeRO-style data parallelism and model state offloading.

The key properties of our new Megatron-DeepSpeed and its design/optimizations released are as follows:

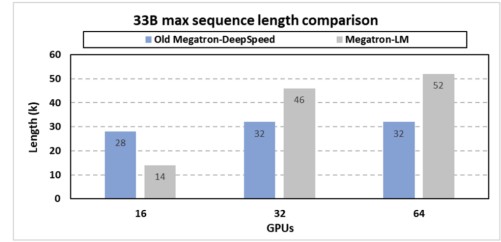

Figure 5: Maximum sequence length support for the 33B GenSLM model.

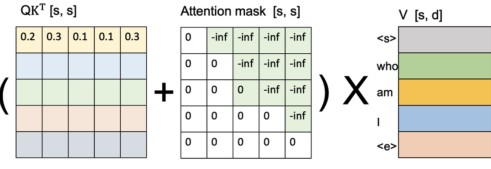

Figure 6: Attention mask operation.

- Enhancing Megatron-style sequence parallelism with our memory optimization techniques for attention mask and position embedding.

- Rotary positional embedding, new fused kernels, and FlashAttention v1 and v2 are also enabled.

- The overall training throughput is improved by up to 2x due to the newly enabled capability of processing larger batch sizes through the new Megatron-DeepSpeed framework.

- An average of 13x longer sequence lengths are achieved compared to the state-of-the-art training frameworks, e.g., enabling training with sequences with over a million tokens.

In the subsequent sections, we will provide a detailed discussion of rebasing efforts/achievements, new Megatron-DeepSpeed core optimizations, experimental evaluation, and comparison analysis against the existing frameworks.

## 3.2 Rebase and Optimizations of Megatron-DeepSpeed Framework

Megatron-DeepSpeed is a framework for training very large-scale LLMs. Since its release, the research community has adopted it for training various LLMs, including the BigScience BLOOM 176B model [10] and Argonne National Lab for GenSLMs. While containing a rich set of optimizations for training LLMs, new features and new demands are coming out rapidly such that having a stable and up-to-date support of Megatron-DeepSpeed is critical for our community of users. For example, there have been more than 1300 new commits on the Megatron-LM side and 75 new commits from the DeepSpeed side since the original Megatron-DeepSpeed release. Therefore, incorporating these new changes and ensuring the robustness of the new framework becomes a fundamental requirement for our science collaborators who use this framework extensively. In this release, we have enabled the following capabilities:

- We integrated several new features, including Megatron-style sequence parallelism, rotary positional embedding, FlashAttention v1 and v2, and new fused kernels from NVIDIA.

- We included additional optimizations specially tailored for long sequence training, such as attention map optimization and position embedding partitioning (discussed next).

- We fixed several conflicts during integration: (1) activation checkpointing where the new fine-grained partial checkpointing technique introduced by Megatron-LM was not compatible with DeepSpeed; (2) model checkpoint save/load when DeepSpeed was used with the newest Megatron-LM; and (3) major refactoring to DeepSpeed pipeline parallelism implementation for GPT models in order to work with the newest Megatron-LM.

- We fully verified the performance and correctness of GPT pretraining after the rebasing. Even though the new Megatron-DeepSpeed has tensor, sequence, and pipeline parallelism, the maximum sequence length is still inadequate. Through profiling, we identified that attention mask and weights of position encoding are main memory bottlenecks.

## 3.3 Further Memory Optimizations in our New Megatron-DeepSpeed

Based on the new rebase, we further enhance the Megatron-style sequence parallelism with our memory optimization techniques for attention mask and position embedding.

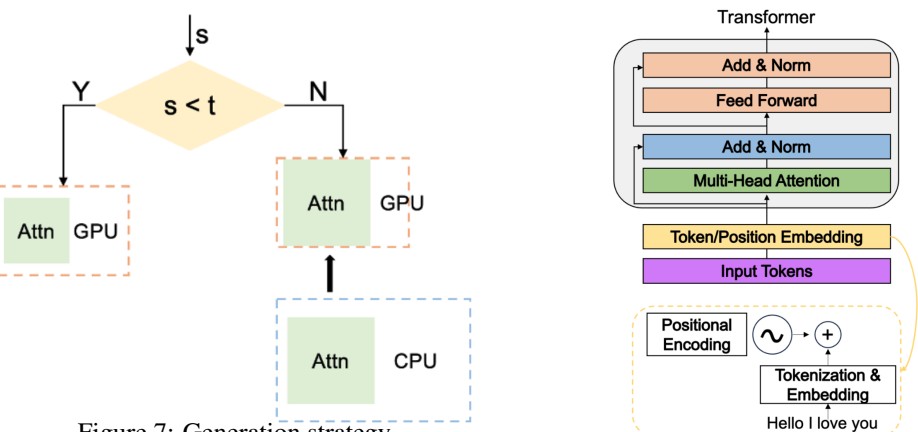

Figure 7: Generation strategy.

Figure 8: Position embedding in Transformers.

### 3.3.1 Memory-Efficient Generation of Attention Masks

Attention mask allows models to only attend to the previous tokens (Figure 6). First, the reason why the attention mask is one of the main memory bottlenecks is because of its size: [s, s], where is the sequence length, making its memory complexity as $O(s^2)$. The size of the attention mask is over 10 GB when the sequence length (s) is larger than 50K (e.g., DNA sequences). Second, PyTorch pre-allocates at least 2X larger GPU memory when generating an attention mask. However, an attention mask is also very important when (1) users explicitly need it when there is no FlashAttention in their virtual environment; and (2) users may want to use customized attention masks to tune their models, not just using casual FlashAttention.

As illustrated in Figure 7, our approach involves initially determining a sequence length threshold through extensive experimentation. This threshold is identified based on achieving optimal system performance while maintaining reasonable memory usage. If the sequence length is below this threshold, we proceed to directly generate an attention mask on the GPU. However, if the sequence length exceeds this threshold, we follow a process in which we initially generate it within CPU memory, perform the necessary operations, and subsequently transfer it to GPU memory. To prevent out-of-memory errors while ensuring consistently high performance, we then establish this threshold based on the underlying GPU hardware (e.g., 16K for A100 40G GPUs).

### 3.3.2 Weights Parallelization of Position Embedding

As shown in Figure 8, position embeddings are used to identify each token's position in the list of tokens. The size of weights of position embedding is [s, d], where s is sequence length and d is the hidden dimension; it is linearly scaled with the sequence length. In the original Megatron-LM's design, each GPU holds a replica of these weights. Training these weights will result in the same size of gradients and m times of the optimizer states (i.e., m is determined by PyTorch). For example, the overall memory consumption is approximately 10 GB per GPU when DNA sequence lengths are longer than 100K.

As shown in Figure 9. Our method is to split weights across all GPUs when enabling sequence parallelism. Each GPU just needs to hold [s/p, d] partial weights. Thus, we reduce GPU memory consumption by p times, where p is the number of GPUs.

### 3.3.3 Algorithmic Support: Relative Position Embedding

Some users may expect a model to achieve extrapolation at inference time for sequences that are longer than it saw during training. We would use relative position embedding [11] (e.g., attention with linear biases) to let users train large language models with shorter sequences, but the trained model can infer much longer sequences.

Table 1: Throughput comparison from the two frameworks on the 33B GenSLM dense model.

| Sequence Length | Old Megatron-DeepSpeed (TFLOPS) | New Megatron-DeepSpeed (TFLOPS) |
|---|---|---|
| 2k | 25 (TP=32) | 68 (TP=32) |
| 4k | 28 (TP=32) | 80 (TP=32) |
| 8k | OOM | 86 (TP=32) |
| 16k | OOM | 92 (TP=32) |
| 32k | OOM | 100 (TP=32) |
| 64k | OOM | 106 (TP=32) |
| 128k | OOM | 119 (TP=32) |
| 256k | OOM | 94 (TP=32) |

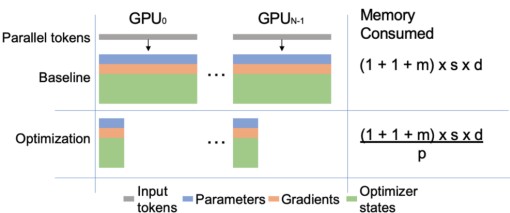

Figure 9: Memory overhead comparison between baseline and the optimized version via parallelizing position embedding.

### 3.4 Performance Improvement of Our New Megatron-DeepSpeed Framework

In order to demonstrate the performance improvement from our new Megatron-DeepSpeed framework, we first show a range of performance comparisons between the old Megatron-DeepSpeed and the New Megatron-DeepSpeed in Table 1, when disabling ZeRO (zero_stage=0). The new Megatron-DeepSpeed is able to support much longer sequence lengths without triggering out-of-memory errors due to (1) Megatron-style sequence parallelism partitions the activation memory when sequence lengths are massive, (2) our enhanced memory optimization through memory-efficient attention mask generation and position embedding parallelization, and (3) FlashAttention V1 and V2 support, which reduces the memory consumption of the attention map calculation from quadratic to linear complexity with respect to the sequence length. The new Megatron-DeepSpeed can achieve higher TFLPOS because it includes new fused kernels from NVIDIA and supports larger batch sizes using our memory optimizations without triggering out-of-memory errors. Appendix A.1 and A.2 present additional evaluation of new Megatron-DeepSpeed framework's capability and scalability. For detailed source code release and tutorial, please visit our release blog for the new Megatron-DeepSpeed.

## 4   Conclusion

We are very proud and excited to announce the DeepSpeed4Science initiative along with several R&D highlights and achievements. We are hosting our new initiative at https://deepspeed4science.ai/, including information about our external colleagues, and current and future DeepSpeed4Science technology releases. One of our high-level goals is to generalize AI system technologies that broadly address the major system pain points for large-scale scientific discoveries. We hope scientists around the world will enjoy the new capabilities unlocked by DeepSpeed4Science through open-sourced software. We are looking forward to better understanding the AI system design challenges that block scientists' discovery progress. We sincerely welcome your participation to help us build a promising AI4Science future.

## Acknowledgment

We thank members of Microsoft DeepSpeed Team for their help on this initial release of Deep-Speed4Science initiative.

**Our Founding Collaborators (in alphabetical order):**

**Argonne National Laboratory:** Rick Stevens, Cristina Negri, Rao Kotamarthi, Venkatram Vishwanath, Arvind Ramanathan, Sam Foreman, Kyle Hippe, Troy Arcomano, Romit Maulik, Maxim Zvyagin, Alexander Brace, Bin Zhang, Cindy Orozco Bohorquez, Austin Clyde, Bharat Kale, Danilo Perez-Rivera, Heng Ma, Carla M. Mann, Michael Irvin, J. Gregory Pauloski, Logan Ward, Valerie Hayot, Murali Emani, Zhen Xie, Diangen Lin, Maulik Shukla, Ian Foster, James J. Davis, Michael E. Papka, Thomas Brettin.

**AMD:** Ashwin Aji, Angela Dalton, Michael Schulte, Karl Schulz.

**Brookhaven National Laboratory:** Adolfy Hoisie, Shinjae Yoo, Yihui Ren.

**Columbia University OpenFold team:** Mohammed AlQuraishi, Gustaf Ahdritz, Christina Floristean.

**Microsoft Research AI4Science team:** Christopher Bishop, Bonnie Kruft, Max Welling, Tie-Yan Liu, Cristian Bodnar, Johannes Brandstetter, Wessel Bruinsma, Chan Cao, Yuan-Jyue Chen, Peggy Dai, Patrick Garvan, Liang He, Elizabeth Heider, PiPi Hu, Peiran Jin, Fusong Ju, Yatao Li, Chang Liu, Ana Lucic, Renqian Luo, Qi Meng, Frank Noe, Paris Perdikaris, Tao Qin, Bin Shao, Yu Shi, Wenlei Shi, Gregor Simm, Megan Stanley, Lixin Sun, Yue Wang, Tong Wang, Zun Wang, Lijun Wu, Yingce Xia, Leo Xia, Shufang Xie, Shuxin Zheng, Jianwei Zhu.

**Microsoft WebXT Weather team:** Pete Luferenko, Divya Kumar, Jonathan Weyn, Ruixiong Zhang, Sylwester Klocek, Volodymyr Vragov.

**NVIDIA:** Yuntian Deng, Weili Nie, Josh Romero, Christian Dallago, Arash Vahdat, Chaowei Xiao, Thomas Gibbs, Anima Anandkumar.

**Oak Ridge National Laboratory:** Prasanna Balaprakash, Gina Tourassi, John Gounley, Heidi Hanson, Thomas E Potok, Massimiliano (Max) Lupo Pasini, Kate Evans, Dan Lu, Dalton Lunga, Junqi Yin, Sajal Dash , Feiyi Wang, Mallikarjun Shankar, Isaac Lyngaas, Xiao Wang, Guojing Cong, Pei Zhang, Ming Fan, Siyan Liu.

**Princeton University:** William Tang, Kyle Felker, Alexey Svyatkovskiy (Microsoft liaison).

**Rutgers University:** Hang Liu.

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

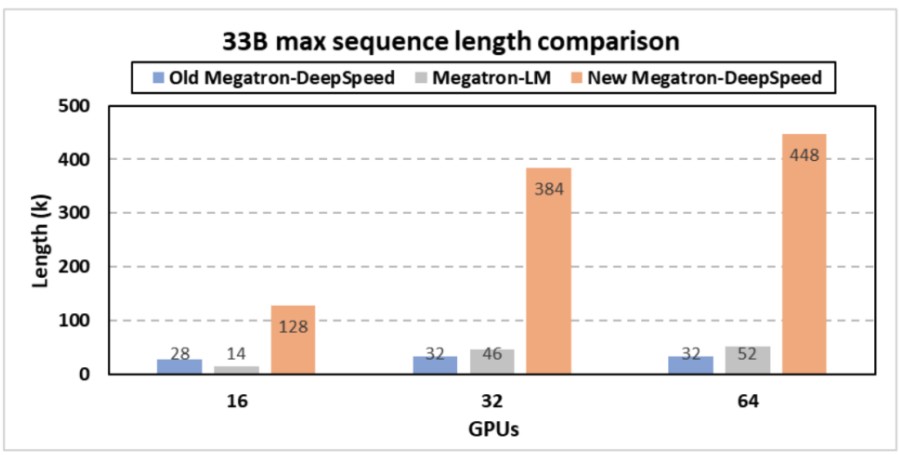

Figure 10: Maximum sequence lengths of 33B GenSLM models supported by different frameworks at different scales. The hardware profiled here are NVIDIA DGX nodes with eight 40G A100 GPUs per node.

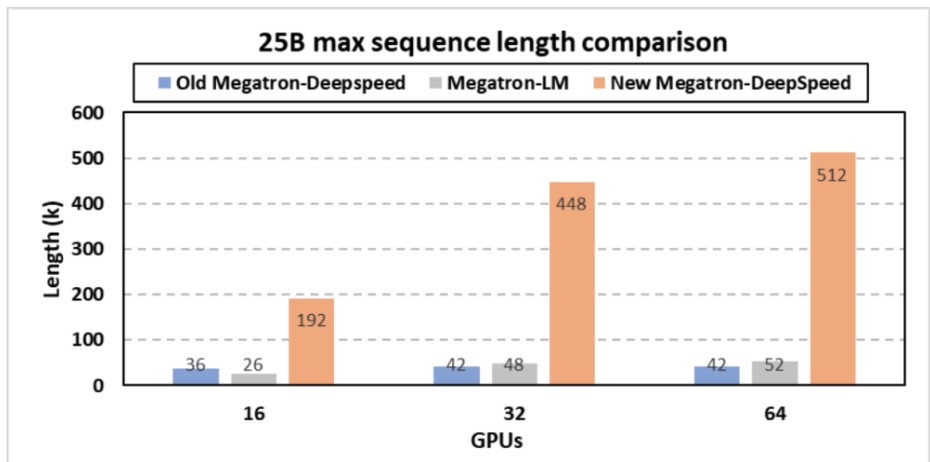

Figure 11: Maximum sequence lengths of 25B GenSLM models supported by different frameworks at different scales. The hardware profiled here are NVIDIA DGX nodes with eight 40G A100 GPUs per node.

## A Appendix

### A.1 New Megatron-DeepSpeed Max Sequence Length Capability

Through our new Megatron-DeepSpeed framework, scientists can now train their large science models like GenSLMs with much longer sequences via a synergetic combination of our newly added memory optimization techniques on attention mask and position embedding, tensor parallelism, pipeline parallelism, sequence parallelism, ZeRO-style data parallelism and model state offloading. Figure 10 and 11 demonstrate that our new framework enables the longest sequence length for GenSLMs' 25B and 33B models by up to 12X and 14X, respectively, over the previous Megatron-DeepSpeed. In terms of supported sequence lengths, this new framework also significantly outperforms NVIDIA's Megatron-LM by up to 9.8X and 9.1X for the 25B and 33B models, respectively. For example, GenSLMs' 25B model can now be trained with a 512K sequence of nucleotides, compared to the Argonne team's original 42K sequence length on 64 GPUs. This drastically improves model quality and scientific discovery scope without additional accuracy loss.

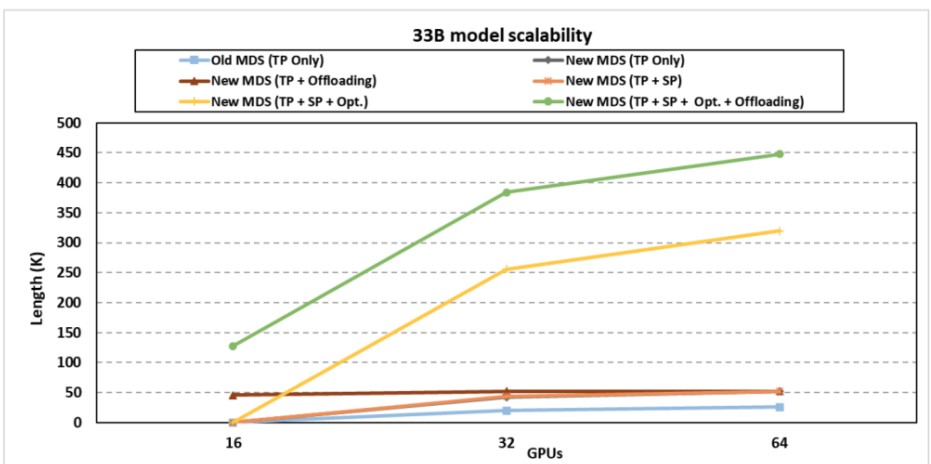

Figure 12: Scalability of the 33B GenSLM model. MDS, TP, SP stand for Megatron-DeepSpeed, tensor parallelism and sequence parallelism.

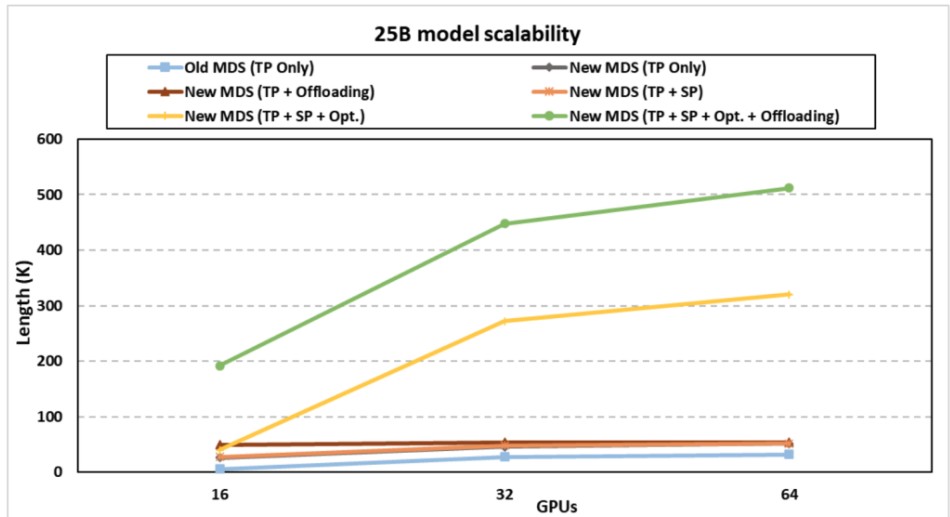

Figure 13: Scalability of the 25B GenSLM model. MDS, TP, SP stand for Megatron-DeepSpeed, tensor parallelism and sequence parallelism.

### A.2 New Megatron-DeepSpeed Scalability Analysis

We further show the scalability of new Megatron-DeepSpeed and what different optimizations entail in Figure 12 and 13. We make two observations. Firstly, when only tensor parallelism and sequence parallelism are used without position embedding optimization, the maximum length of the sequence this training system can support is about 50K, and continuing to increase GPUs will not allow the system to support longer sequences. Secondly, when enabling sequence parallelism, the maximum length of the sequence that can be supported varies within 4K.

There are several reasons behind these observations. Firstly, during training, the majority of a device's memory is used for model state when the sequence length is small. However, as the sequence length increases, the activation memory and temporary buffers can grow significantly. For instance, GPT-style models require O(seq_length x n_layer x hidden_dim x batch_size) to store activations, and O(seq_length x seq_length) to store the attention map, and O(3 x seq_length x hidden_dim ) to train the position embedding.

Secondly, the attention map is linearly proportional to the sequence length, while the latter has quadratic memory complexity. For a 25B parameter GPT model trained with a sequence length of 100K and a batch size of 1, the activation memory requires about 12 GB and the attention map requires at least 10 GB per device, both of which are non-trivial. By using techniques such as model parallelism, we can reduce the memory footprint by using aggregated device memory for activation

memory from 480 GB to 12 GB. Finally, we also optimized the attention map's memory usage by avoiding allocating temporary buffers on the device, which reduces the peak memory consumption from 54 GB (Out of Memory) to 39 GB. Even if we only use casual flash attention (avoid generating attention map explicitly), the memory requirement for training position embedding is linearly scaled with sequence length, and the needed memory is over 10 GB when a sequence length is over 100K.

