# OpenReview forum: "DeepSpeed4Science Initiative: Enabling Large-Scale Scientific Discovery through Sophisticated AI System Technologies"
_NeurIPS.cc/2023/Workshop/AI4Science — NeurIPS2023-AI4Science Poster_

### Official Review · Reviewer_p9Qn · 2023-10-22
**Questions of Generality, Tone and Discovery**

**Rating:** 5
**Confidence:** 2

**Review:**

This manuscript introduces a new PyTorch library designed for AI applications in biology, offering what seems to be promising technical advantages, including superior memory efficiency. Note: I'd like to mention that I am not a biologist and thus, I'm not a domain expert on the specific area showcased in this paper. Consequently, my comments should be taken as those of an interested reader from a distant field who uses AI for discovery.

The primary contribution appears to be the presentation of a novel computational ML platform, Deepspeed4Science. This platform is derived from its predecessor, the DeepSpeed library by Microsoft, which is known for enhancing PyTorch with advanced optimizations like ZeRO and model parallelism. Deepspeed4Science aims to tailor these capabilities specifically for scientific tasks.

However, there are areas where the manuscript could provide clearer demonstrations and better support for its claims:

Key Observations:

1) Clarity in Introduction and Goals: The abstract, introduction, and conclusion are somewhat nebulous, making it difficult to identify the specific contributions and goals of this work. Broad statements like "scientists around the world will enjoy the new capability" or "major system pain points for large-scale scientific discoveries" are not backed with evidence.

2) Scope and Generality: The manuscript suggests that Deepspeed4Science could be applicable across various scientific domains. Yet, the evidence is limited to two examples from biology. The potential utility in other domains such as (quantum) chemistry or physics is not demonstrated and not obvious to the reviewer.

3) Scientific Discoveries: The authors emphasize the potential of Deepspeed4Science for scientific discoveries (with a bold claim already in the title), but the provided examples seem to be areas already addressed by other methods. It remains unclear from the manuscript whether and how novel discoveries can be made.

Suggestions:

- A more measured tone might be beneficial, avoiding phrases like "proud to present," which can come off as promotional rather than strictly scientific.

- If the aim is to highlight potential rather than realized benefits, this distinction should be clearer. Instead of broad claims, the manuscript might delve deeper into what sets Deepspeed4Science apart and its potential advantages.

In conclusion, while the manuscript introduces a new framework, Deepspeed4Science, its broader relevance and effectiveness for AI in various scientific fields remain to be seen.